# Captioning and Task-Specific Prompting for Improved VLM Performance

## Abstract

Vision-language models (VLMs) have transformed tasks requiring visual and reasoning abilities, such as image retrieval and visual question answering (VQA). Despite their success, VLMs face significant challenges with tasks involving geometric reasoning, algebraic problem-solving, and counting. These limitations stem from difficulties in effectively integrating multiple modalities and accurately interpreting such tasks. We propose an efficient, question-driven image captioning pipeline to enhance visual question answering abilities in mathematical contexts. Our method extracts keywords from the question, generates targeted captions for each image-question pair using those keywords, and uses the caption as a prompt for QnA. We propose utilizing task-specific guidance as an "approach" to enhance the VQA and captioning process. Additionally, we evaluate the robustness of these models against adversarial prompts to ensure that our captioning-based approach does not compromise much on robustness. Our pipeline is tested on diverse math-related and visual reasoning tasks across multiple datasets and VLMs.

## Introduction

VLMs have made substantial progress in advancing multimodal learning. Models such as CLIP (Radford et al. 2021), BLIP (Li et al. 2022), and ImageBind (Girdhar et al. 2023) offer powerful encoders with excellent representation capabilities, demonstrating high transferability in recognition and understanding tasks. Additionally, VLMs like LLaMA Adapter (Zhang et al. 2023), BLIP2 (Li et al. 2023a), Flamingo (Alayrac et al. 2022), and LLaVA (Liu et al. 2024) combine visual encoders from discriminative VLMs with Large Language Models (LLMs) like LLaMA (Touvron et al. 2023) and GPT-3 (Brown 2020), capitalizing on LLMs' exceptional generative abilities. Recent models, including GPT-4 Vision (Achiam et al. 2023), Claude 3.5 Sonnet (AI 2024), and Gemini (Team et al. 2024), have achieved human-level performance across various benchmarks such as, Mmmu (Yue et al. 2024), MathVista (Lu et al. 2023), DocVQA (Mathew, Karatzas, and Jawahar 2021). While these models perform well on standard benchmarks, studies indicate that they often struggle to fully comprehend the visual aspects of mathematical tasks, particularly failing to capture relational and spatial information (Chen et al. 2024) within diagrams and relying heavily on textual cues instead (Zhang et al. 2024b). Various works have further shown that

these models lack both spatial scene understanding (Tong et al. 2024) and temporal awareness (Li et al. 2023b), with some studies even revealing difficulties with basic visual tasks that humans handle effortlessly. This underscores the need to enhance not only the visual reasoning and understanding capabilities of VLMs but also their fundamental ability to perceive and interpret images effectively (Rahmanzadehgervi et al. 2024). In this work, we evaluate the visual capabilities of these models on math-related tasks, specifically testing their ability to truly 'see' and interpret visual information. We propose a zero-shot pipeline that uses task-specific guidance to generate image captions to improve performance on these tasks. Additionally, we test the model's robustness against adversarial prompts framed as incorrect yet relevant problem-solving strategies. Given that VLMs are largely pre-trained for caption generation (Yang et al. 2024a), (Ramos et al. 2023) and many downstream tasks (Zhang et al. 2024a) require textual outputs, we hypothesize that guiding the model to create detailed captions encourages it to focus more on the visual content, reducing its reliance on textual cues from the question alone. Captioning, thus, enables the model to capture all essential image attributes, which is especially critical for mathematical reasoning.

## Background and Related work

Various prior works have tried to improve the performance of VLMs without requiring external fine-tuning, techniques such as in-context learning(Sarch et al. 2024; Nulli et al. 2024; Doveh et al. 2024; Sun et al. 2024; Zhao et al. 2023),chain of thought prompting(Zhang et al. 2024c; Ge et al. 2023), few-shot prompting (Brown et al. 2020; Chen et al. 2023), have enhanced VLM performance, yet these models continue to face challenges in mathematical tasks, particularly in counting, leading some researchers to describe them as "blind" to visual information (Rahmanzadehgervi et al. 2024). While research suggests that much of the reasoning displayed by Vision-Language Models (VLMs) may rely more on the phrasing of questions than on the images themselves (Zhang et al. 2024b). This limitation is especially pronounced in tasks that depend heavily on visual details, such as counting nested shapes, identifying line intersections, identifying the number of objects in an image, and solving geometrical problems where VLMs con-

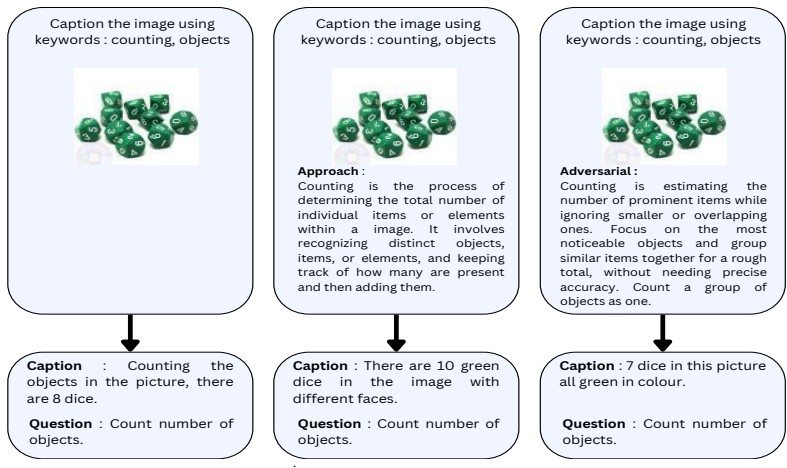

Figure 1: Example of our Captioning Approach

sistently fall short. To evaluate these visual abilities, specialized datasets such as Math Vision (Wang et al. 2024), Count Bench (Paiss et al. 2023), Blind (Rahmanzadehgervi et al. 2024), and Geo170k(Gao et al. 2023a) have been introduced. In efforts to improve general visual question answering (VQA) performance, various techniques have been developed, including question-driven image captions (Özdemir and Akagündüz 2024) that are processed by language models. These methods have shown promise, particularly in enhancing direct visual question-answering tasks. Further, some studies aim to improve the reasoning abilities of VLMs by directly fine-tuning them for tasks such as question answering (Deng et al. 2024; Roberts and Roberts 2024) and mathematical problem-solving (Yang et al. 2024b). In contrast, our research focuses on enhancing the visual performance of VLMs through a pipeline that leverages simple prompting techniques, eliminating the need for fine-tuning and thereby reducing computational overhead.

## Methodology

We evaluated Vision-Language Models (VLMs) on a range of visual tasks to assess their visual abilities. To ensure robustness and generalizability, we selected tasks from four datasets covering various question types, including geometry, counting, algebra, and mathematical reasoning. Our experiments employed a diverse set of four VLMs to evaluate the generalization of our approach. Each model was tested across four distinct methods to comprehensively assess our proposed approach:

1. **Vanilla QnA:** We used a classical zero-shot approach where each model was directly queried with questions related to images from the datasets. This formed the baseline of our experiments.

2. **Approach-based QnA:** We used Gemini-1.5 (Team et al. 2024) to provide the approach to solve the desired question by simply giving it the question and asking a comprehensive approach for the question. Then we prompted the VLMs being tested to solve the question by

providing the approach with it.

3. **Vanilla Keyword Captioning:** We used the base model to create captions for the image, guiding the captioning process with specific keywords related to the question. These keywords were obtained from the Llama 3.1-Instruct model[1] (Dubey et al. 2024) by prompting the model to provide 3-5 concise keywords for each question. After generating the captions with the VLM, using the keywords as context, we input them into an LLM(Gemini) and asked the relevant questions for each task.

4. **Approach-based Captioning:** We conducted additional tests to evaluate the impact of providing more context in prompts for generating captions. First, we fed the question to Gemini, asking it to develop an approach to solve the problem. Following this, we fed the proposed approach, along with the image and relevant keywords generated from Llama to the VLM. We asked the VLM to caption the image using the keywords and approach as context. Finally, we input this caption into an LLM(Gemini) and asked the model to answer the required question based on the caption.

Further, to assess the robustness of our proposed approach, we tested the models across the following 2 tasks as well:

1. **Adverserial-based QnA:** We appended an adversarial prompt, generated using Gemini, to the question. This contained misleading information about the approach to be followed to solve the question. We then queried each VLM on images in the dataset, using the question appended with the incorrect adversarial approach.

2. **Adverserial-based Captioning:** Similar to approach-based captioning, we generated adversarial approaches using Gemini by asking it to generate a "wrong" approach to solve the given question. Then, we generated the image caption from the VLMs with the adversarial

---

[1]https://huggingface.co/meta-llama/Llama-3.1-8B-Instructg

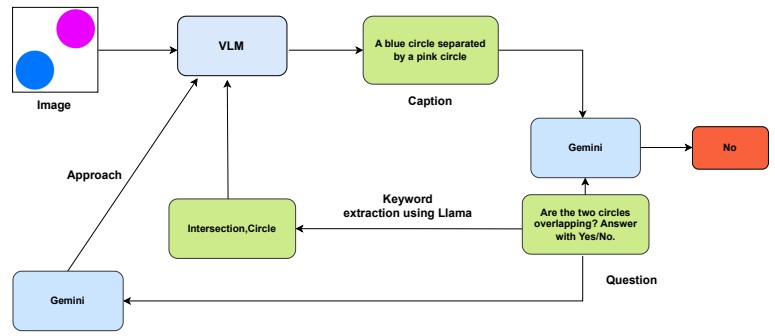

Figure 2: Example of our Caption Based Approach

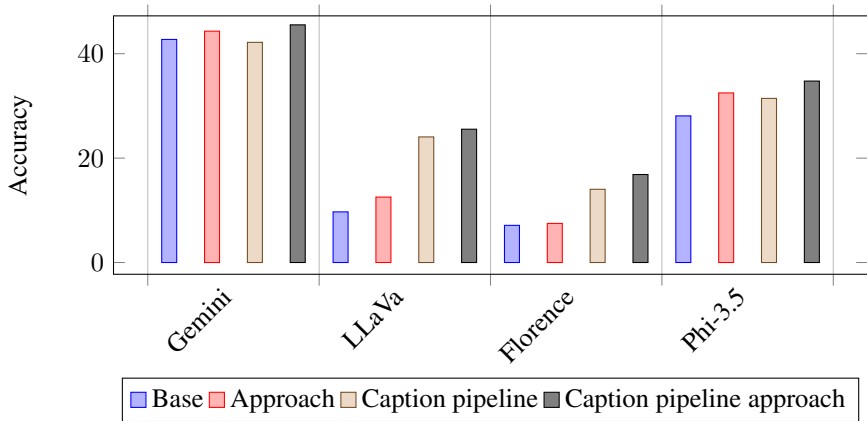

Figure 3: Comparison of different Methods

approach along with the image as inputs. Finally, we used an LLM (Gemini-1.5) and asked it to answer the required question based on the caption.

## Experiments

We chose diverse models and techniques to prove and test our hypothesis. We chose four datasets, Geo170k (Gao et al. 2023b), CountBench (Sindagi, Yasarla, and Patel 2020), Blind (Rahmanzadehgervi et al. 2024), and MathVision (Wang et al. 2024), containing various tasks related to geometry, reasoning, algebra, and counting. Also, we split the MathVision dataset into three subparts: mainly vision-based, geometry-based, and mathematics-based. To cover a range of model sizes, we selected a diverse set of open-source models, namely Gemini-1.5-Flash (Team et al. 2024), LLaVa (Liu et al. 2024), Florence-2 (Xiao et al. 2024), and Phi 3.5 Vision Instruct (Abdin et al. 2024). This selection ensures a variety of model sizes, from smaller ones with fewer parameters to larger, more complex models.

- **LLaVa:** For LLaVa, we used the GroqAPI [2] to access the model.
- **Gemini-1.5-Flash:** For Gemini, we used Google AI studio[3] to access the model.

---

[2]https://console.groq.com/

[3]https://aistudio.google.com/

- **Florence-2:**[4][5] For Florence-2, we used the open-sourced model available on huggingface .
- **Phi 3.5 Vision Instruct:**[6] we used the open-sourced model available on huggingface.

**Note:** For Florence-2 captions were generated using the token <DETAILED CAPTION>, and the approach was passed onto the detailed caption. For other models, the approach was passed during the caption generation stage. Additionally, the Florence-2 direct checkpoint was unable to perform QnA-related tasks, so we used Florence-2 DocVQA for QnA-related tasks.

## Results

Vanilla captioning leads to overall performance improvements, though its effectiveness varies with model size—it tends to benefit smaller models more than larger ones. In fact, Gemini registers a small performance drop using vanilla captioning. Approach-based QnA demonstrates results above the baseline, supporting adding additional information and context as prompts. Using task-based prompts for generating captions consistently leads to significant improvements across all models and datasets. This approach

---

[4]https://huggingface.co/microsoft/Florence-2-large

[5]https://huggingface.co/HuggingFaceM4/Florence-2-DocVQA

[6]https://huggingface.co/microsoft/Phi-3.5-vision-instruct

| Model | Base | Approach | Adv. | Caption | Caption Approach | Caption Adv. |
|---|---|---|---|---|---|---|
| **Gemini-1.5-Flash** | 42.73 | 44.32 | 43.86 | 42.18 | **45.53** | 37.70 |
| **LLaVa** | 09.70 | 12.54 | 9.61 | 24.06 | **25.54** | 23.15 |
| **Florence-2** | 07.12 | 07.49 | 02.89 | 14.03 | **16.86** | 14.56 |
| **Phi-3.5-Vision** | 28.09 | 32.49 | 28.92 | 31.44 | **34.76** | 29.27 |

Table 1: Model-wise comparison of the accuracy of our different approaches

| Dataset | Base | Approach | Adv. | Caption | Caption Approach | Caption Adv. |
|---|---|---|---|---|---|---|
| **Math-vision** | 10.13 | 14.06 | 12.31 | 19.28 | **22.31** | 21.43 |
| **CountBench** | 32.58 | 33.33 | 34.50 | 27.16 | **34.16** | 28.00 |
| **Geo** | 16.22 | 20.46 | 14.67 | 24.00 | **28.56** | 26.00 |
| **Blind** | 31.52 | 32.12 | 25.98 | 31.94 | **32.81** | 28.60 |

Table 2: Dataset-wise comparison of accuracy of the different approaches

demonstrates the greatest improvement in performance, making it a promising method. We also observe a performance drop with adversarial prompting, though this drop is less, thus showing that captioning does not compromise on robustness. Performance differences across datasets are notable, too (Table 4). While average model performance remains relatively unchanged on the CountBench dataset, which focuses on counting tasks, captioning yields larger accuracy improvements on the Geo and MathVision datasets, which consist of geometry and visual mathematics tasks. This aligns with our hypothesis that captioning allows the model to focus more on visual cues necessary for answering diagram-related questions (Zhang et al. 2024b).

## Conclusion

The results of our study support our initial hypothesis: Vision-Language Models (VLMs) exhibit significant limitations in mathematical and visual tasks, especially those involving numbers and counting—tasks that test the models' ability to "see" rather than merely reason. While captioning with task-specific keywords can improve performance in certain cases by directing the models' focus toward visual details, our findings reveal that its effectiveness depends on factors such as dataset type, task complexity, and model size. Larger models, particularly those pre-trained on similar tasks, tend to perform better, underscoring the role of pre-training, as seen in Gemini, a trend less evident in smaller models. Our experiments demonstrate that we can improve the mathematical and visual capabilities of Vision-Language Models (VLMs), resulting in enhanced performance. When using adversarial prompting, we observed a consistent performance drop across all models. However, this drop was moderate and, on average, similar to the decline seen in adversarial QnA tasks, indicating that captioning maintains a satisfactory level of robustness. In summary, approach-based captioning boosts the reasoning abilities of VLMs and increases their focus on visual cues, thereby reducing their reliance on textual input. This approach offers a promising direction for addressing VLMs' limitations in perceiving numerical information and handling visually complex mathematical tasks. By using structured prompts derived directly from the question and incorporating reasoning-guided methods, we can mitigate some of these weaknesses and improve VLM performance in such problem-solving scenarios.

## Limitations and Future Work

Testing the reasoning capabilities of multimodal models is a broad research area, and we propose ways to enhance model generalizability in our study. Due to resource constraints, we couldn't experiment with large-scale models like GPT-4 (Achiam et al. 2023) or use full-size datasets, limiting our ability to explore scalability fully. Future work with greater resources could expand on robustness and scalability across diverse model architectures, including models like Claude Sonnet 3.5 (cla 2024) and GPT-4 (Achiam et al. 2023).

Our current approach involves using an LLM to process captions from the VLM for answer extraction. Simpler parsing techniques might be sufficient, given the quality of generated captions. Additionally, advanced methods, such as sophisticated prompt engineering as those used with LLMs (Agrawal et al. 2024; Zhou et al. 2022; Wang and Zhou 2024; Yao et al. 2024), or incorporating domain-specific knowledge (Liu et al. 2025) followed by fine-tuning for more precise, contextually relevant captions could lead to a higher performance increase. Our study also doesn't prioritize interpretability, which could be addressed by examining how captioning influences the interaction between visual and textual inputs in Vision-Language Models (VLMs) and impacts performance. Exploring interpretable alternatives, like graph networks, could further improve transparency and deepen understanding of the models' reasoning processes.

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

| Model | Base | Random | Caption | Caption Random |
|---|---|---|---|---|
| **Gemini-1.5-Flash** | 42.73 | 43.86 | 42.18 | 38.62 |
| **LLaVa** | 09.70 | 10.61 | 24.06 | 25.07 |
| **Florence-2** | 07.12 | 02.89 | 14.02 | 16.35 |
| **Phi-3.5-Vision** | 28.09 | 28.91 | 31.44 | 28.71 |

Table 3: Model-wise comparison of the accuracy of random approach

| Dataset | Base | Random | Caption Approach | Caption Random |
|---|---|---|---|---|
| **Math-vision** | 10.13 | 13.31 | 19.27 | 19.61 |
| **CountBench** | 32.58 | 34.5 | 27.16 | 31 |
| **Geo** | 16.22 | 14.66 | 24.00 | 23.00 |
| **Blind** | 31.52 | 25.98 | 31.94 | 31.90 |

Table 4: Dataset-wise comparison of the accuracy of our different approaches

## Datasets

The following datasets were used for our experiments:

- **Math Vision:**[7] The Math Vision dataset is a curated collection of 3,040 high-quality mathematical problems with visual contexts from real math competitions. For our experiments, we broadly divided the dataset into three categories:

  - **Visual Based:** This was originally split into Area, Angles, and Length-related tasks.
  - **Geometry Based:** This was originally split into categories: Analytical Geometry, Combinatorial Geometry, Transformation Geometry, Descriptive Geometry and Solid Geometry.
  - **General Mathematics:** This was originally split into categories: Graph Theory, Logic, Algebra, Combinatorics, Statistics, and Arithmetic.

- **Blind:**[8] The Blind dataset consisted of images and question-answer pairs about visual tasks. We used a subset of 150 images per task. The tasks include counting the number of intersections of 2 circles or lines, checking if 2 lines are intersecting, counting the number of rows and columns in a grid, finding the number of overlapping circles in an image, and finding the number of paths between 2 points in a subway connection image.

- **Countbench:**[9] The CountBench dataset contained a total of 540 images containing between two and ten instances of a particular object, where their corresponding captions reflect this number. This dataset is a benchmark dataset for counting related tasks.

- **Geo170k:**[10] The Geo dataset contained more than 170K geometric image-caption and question-answer pairs. We used a subset of 500 images to conduct our experiments.

---

[7]https://huggingface.co/datasets/MathLLMs/MathVision
[8]https://huggingface.co/datasets/XAI/vlmsareblind
[9]https://huggingface.co/datasets/nielsr/countbench
[10]https://huggingface.co/datasets/Luckyjhg/Geo170K

## Shortcomings

One notable limitation of our pipeline is that the model's strategy for addressing the question is derived directly from the question itself. Specifically, we generate the approach using an LLM (Gemini), which means that the effectiveness of the pipeline is influenced by both the quality of the questions in the dataset and the nature of the task being tackled. This was evident in our main paper, where we observed a modest improvement on tasks like count-bench and blind tasks but saw a significant boost in performance on the Math-vision and Geo datasets. Additionally, using an LLM to generate the task approach inherently contributes to the performance gains of our pipeline compared to the baseline. We hypothesize that utilizing more advanced LLMs, such as GPT-4 or Claude, would further enhance these improvements. However, we were unable to evaluate this due to computational and financial constraints. Future research could focus on developing a standard benchmark dataset or exploring how much the complexity of the LLM influences the performance of such a pipeline used to generate the task approach.

## Random Approach

In our experiment, we also explored random-approach-based captioning. This method involved providing the VLM with the image, an LLM-generated random phrase, and relevant keywords to create a detailed caption. However, the results were highly inconsistent, showing significant variability in performance(Table4). In several instances, this approach outperformed the baseline, but we could not determine the underlying cause of this improvement, although we hypothesize that the model tends to disregard random information and in doing so becomes more cautious and robust. Given these unpredictable outcomes, further investigation into this approach could be a potential direction for future research.

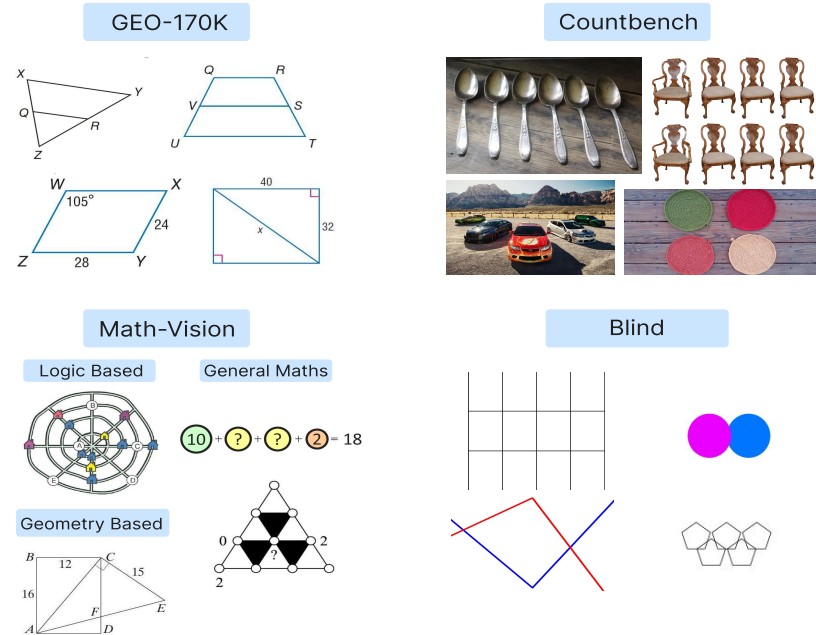

Figure 4: Example of images in the datasets

| Dataset name | Task in dataset | Keywords for task |
| --- | --- | --- |
| MathVision | Analytic Geometry | analytic geometry |
| | Algebra | algebra, mathematics, logic |
| | Transformation Geo | transformation, geometry |
| | Statistics | statistics, graph |
| | Angle | metric geometry, angles, mathematics, logic |
| | Combinatorics | combinatorics, logic |
| | Descriptive Geo | descriptive geometry, mathematics |
| | Logic | logics, reasonings |
| | Length | lengths, geometry |
| | Arithmetic | arithmetic, logics, mathematics |
| | Area | area, geometry |
| | Combinatorial Geo | combinatorial, geometry |
| | Solid Geometry | solids, geometry |
| | Graph Theory | logics, connections, graphs |
| Countbench | Counting objects | counting, objects |
| GEO170K | Geometry problems | geometry, mathematics |
| BLIND | Line intersection | count number, intersections |
| | Two line intersection | lines, intersecting |
| | Interior pentagon | count, number, pentagons |
| | Subway | subway lines, count, paths |
| | Rows/columns | count, rows, columns |
| | Two circles | circles, touching |

Table 5: Dataset Information and Keywords

| Question | Approach |
|---|---|
| Are the two circles overlapping? | Measure the distance between their centers and compare it to the sum of their radii. If the distance equals the sum of the radii, the circles are touching. If the distance is greater than the sum of the radii, the circles are apart. |
| Count the pentagons in the image. | Analyze the image containing multiple pentagons. Identify and count each distinct pentagon, regardless of size or position. Account for all visible pentagons, including those partially obscured. |
| Count the number of rows and columns. | Examine the grid structure by locating horizontal lines (rows) and vertical lines (columns). Count the total number of rows and columns, including visible ones without content. |
| What will be the fourteenth animal the magician pulls out of his hat? | Identify the five-animal sequence pattern. Determine where the fourteenth animal falls within the repeated pattern. |
| Which number do you have to write in the last daisy? | Identify the pattern in the numbers within each daisy. Determine how the pattern changes from one daisy to the next. Apply this pattern to the final daisy. |
| For how many minutes are there exactly two lights on at the same time? | Analyze the image to identify time intervals where exactly two lights are on simultaneously. Calculate the total duration of these intervals. |
| Find UT in trapezoid QRTU. | Use the relationship between the midpoints of the legs and the bases of a trapezoid. The line segment connecting the midpoints is parallel to the bases and its length is the average of the base lengths. |
| Use parallelogram to find x. | Utilize the properties of parallelograms, specifically the relationship between opposite sides and angles, to set up and solve an equation involving 'x'. |
| Find JK in intersecting chords. | Apply the Intersecting Chords Theorem: the product of the segments of one chord equals the product of the segments of the other chord. Set up and solve the equation. |
| Find GF in trapezoid CDFG with median HE. | Use the property that the median of a trapezoid is parallel to the bases and half the sum of their lengths. Express GF in terms of HE and CD, then solve. |

Table 6: Some Questions and Their Generated Approaches

| Task | 0-Shot | Approach | Adv-App | Caption | App-Capt | App-Capt-Adv |
|---|---|---|---|---|---|---|
| **MathVision** | | | | | | |
| Angles | 36 | 36 | 24 | 41 | 33 | 34 |
| Area | 27 | 28 | 24 | 37 | 39 | 28 |
| Length | 30 | 32 | 24 | 35 | 29 | 32 |
| Descriptive Geo | 34 | 26 | 28 | 19 | 24 | 24 |
| Analytic Geo | 16 | 18 | 22 | 25 | 25 | 20 |
| Combinatorial Geo | 22 | 26 | 19 | 23 | 24 | 10 |
| Transformation Geo | 18 | 24 | 20 | 25 | 17 | 28 |
| Solid Geo | 20 | 32 | 20 | 21 | 30 | 16 |
| Graph Theory | 28 | 24 | 22 | 29 | 29 | 22 |
| Arithmetic | 26 | 26 | 28 | 27 | 38 | 26 |
| Logic | 32 | 32 | 36 | 22 | 39 | 22 |
| Combinatorics | 20 | 20 | 12 | 27 | 31 | 14 |
| Algebra | 26 | 22 | 30 | 23 | 35 | 28 |
| Statistics | 26 | 24 | 24 | 27 | 28 | 22 |
| **Dataset Average** | 26.44 | 27.29 | 23.38 | 28.89 | 29.31 | 24.42 |
| **GEO170K** | | | | | | |
| Geometry problems | 30 | 33.33 | 32 | 35 | 39 | 34 |
| **CountBench** | | | | | | |
| Counting objects | 62 | 68 | 64.67 | 49 | 64.66 | 52 |
| **Blind** | | | | | | |
| Line Intersection | 50 | 58 | 44 | 48 | 46 | 35 |
| Two line intersection | 72 | 70 | 54 | 78 | 78 | 70 |
| pentagon | 53 | 42 | 24 | 49 | 44 | 43.34 |
| Subway | 23 | 23 | 26 | 0 | 0 | 0 |
| Rows/columns | 28 | 32 | 31 | 24 | 26 | 11 |
| Two circles | 89 | 67 | 92 | 86 | 85 | 83 |
| **Dataset Average** | 52.50 | 48.67 | 45.16 | 46.87 | 47.16 | 40.39 |

Table 7: Gemini Model Performance Across Various Tasks

| Task | 0-Shot | Approach | Adv-App | Caption | App-Capt | App-Capt-Adv |
|---|---|---|---|---|---|---|
| **MathVision** | | | | | | |
| Angles | 6 | 14 | 8 | 38 | 34 | 28 |
| Area | 2 | 4 | 2 | 24 | 26 | 26 |
| Length | 6 | 14 | 14 | 20 | 26 | 28 |
| Descriptive Geo | 20 | 12 | 16 | 20 | 22 | 24 |
| Analytic Geo | 4 | 12 | 8 | 22 | 12 | 26 |
| Combinatorial Geo | 12 | 14 | 12 | 16 | 12 | 8 |
| Transformation Geo | 18 | 16 | 14 | 24 | 24 | 28 |
| Solid Geo | 4 | 10 | 6 | 22 | 26 | 20 |
| Graph Theory | 12 | 6 | 6 | 18 | 24 | 28 |
| Arithmetic | 2 | 14 | 10 | 16 | 22 | 14 |
| Logic | 10 | 16 | 8 | 16 | 24 | 18 |
| Combinatorics | 4 | 2 | 8 | 16 | 14 | 20 |
| Algebra | 0 | 4 | 4 | 26 | 28 | 16 |
| Statistics | 6 | 12 | 6 | 20 | 12 | 14 |
| **Dataset Average** | 7.31 | 10.82 | 8.73 | 22.27 | 22.84 | 22.29 |
| **GEO170K** | | | | | | |
| Geometry problems | 5 | 8 | 6.67 | 33.33 | 36 | 32.66 |
| **CountBench** | | | | | | |
| Counting objects | 12 | 14 | 10 | 14.66 | 16 | 14 |
| **Blind** | | | | | | |
| Line Intersection | 4 | 1 | 3 | 6 | 5 | 6 |
| Two line intersection | 32 | 42 | 30 | 64 | 68 | 55 |
| pentagon | 2 | 3 | 2 | 11 | 10 | 9 |
| Subway | 6 | 8 | 7 | 28 | 25 | 26 |
| Rows/columns | 5 | 9 | 7 | 6 | 10 | 6 |
| Two circles | 38 | 41 | 36 | 41 | 46 | 40 |
| **Dataset Average** | 14.50 | 17.33 | 14.16 | 26 | 27.33 | 23.67 |

Table 8: LLAVA Model Performance Across Various Tasks

| Task | 0-Shot | Approach | Adv-App | Caption | App-Capt | App-Capt-Adv |
|---|---|---|---|---|---|---|
| **MathVision** | | | | | | |
| Angles | 5 | 2 | 0 | 22 | 24 | 17 |
| Area | 4 | 3 | 0 | 22 | 26 | 24 |
| Length | 6 | 4 | 0 | 16 | 26 | 25 |
| Descriptive Geo | 3 | 3 | 1 | 16 | 24 | 0 |
| Analytic Geo | 2 | 5 | 2 | 17 | 16 | 20 |
| Combinatorial Geo | 3 | 2 | 1 | 19 | 15 | 8 |
| Transformation Geo | 1 | 2 | 1 | 20 | 23 | 18 |
| Solid Geo | 4 | 3 | 0 | 19 | 21 | 19 |
| Graph Theory | 1 | 2 | 0 | 17 | 23 | 28 |
| Arithmetic | 2 | 3 | 1 | 15 | 23 | 18 |
| Logic | 2 | 2 | 1 | 15 | 24 | 20 |
| Combinatorics | 2 | 4 | 1 | 15 | 15 | 17 |
| Algebra | 2 | 3 | 0 | 24 | 25 | 18 |
| Statistics | 1 | 1 | 1 | 18 | 16 | 16 |
| **Dataset Average** | 3.09 | 2.83 | 0.56 | 18.51 | 22.04 | 19.25 |
| **GEO170K** | | | | | | |
| Geometry problems | 0 | 1.34 | 0 | 4 | 9 | 6 |
| **CountBench** | | | | | | |
| Counting objects | 3 | 4 | 2 | 6 | 8 | 8 |
| **Blind** | | | | | | |
| Line Intersection | 15 | 10 | 31 | 37 | 38 | 30 |
| Two line intersection | 74 | 76 | 31 | 69 | 70 | 69 |
| pentagon | 0 | 0 | 0 | 0 | 0 | 0 |
| Rows/columns | 0 | 0 | 0 | 0 | 0 | 0 |
| Two circles | 23 | 23 | 23 | 32 | 34 | 26 |
| **Dataset Average** | 22.4 | 21.8 | 17 | 27.6 | 28.4 | 25 |

Table 9: Florance-2 Model Performance Across Various Tasks

| Task | 0-Shot | Approach | Adv-App | Caption | App-Capt | App-Capt-Adv |
|---|---|---|---|---|---|---|
| **MathVision** | | | | | | |
| Angles | 0 | 12 | 18 | 10 | 20 | 18 |
| Area | 2 | 18 | 16 | 14 | 20 | 18 |
| Length | 6 | 22 | 14 | 14 | 24 | 18 |
| Descriptive Geo | 6 | 16 | 20 | 16 | 22 | 22 |
| Analytic Geo | 2 | 10 | 14 | 14 | 14 | 26 |
| Combinatorial Geo | 4 | 18 | 10 | 12 | 12 | 8 |
| Transformation Geo | 4 | 18 | 24 | 14 | 10 | 8 |
| Solid Geo | 6 | 14 | 12 | 4 | 6 | 4 |
| Graph Theory | 0 | 16 | 14 | 10 | 16 | 14 |
| Arithmetic | 8 | 20 | 16 | 20 | 16 | 18 |
| Logic | 6 | 12 | 14 | 10 | 22 | 18 |
| Combinatorics | 6 | 14 | 12 | 14 | 14 | 14 |
| Algebra | 2 | 12 | 18 | 4 | 10 | 12 |
| Statistics | 2 | 6 | 4 | 18 | 24 | 18 |
| **Dataset Average** | 3.68 | 15.28 | 15 | 12.44 | 17.04 | 15.75 |
| **GEO170K** | | | | | | |
| Geometry problems | 18.67 | 26.67 | 25.33 | 38 | 40.67 | 38 |
| **CountBench** | | | | | | |
| Counting objects | 53.33 | 47.33 | 50 | 45 | 50 | 38 |
| **Blind** | | | | | | |
| Line Intersection | 35 | 39 | 32 | 33 | 33 | 31 |
| Two line intersection | 59 | 62 | 55 | 70 | 69 | 54 |
| pentagon | 12 | 13 | 2 | 6 | 9 | 4 |
| Subway | 22 | 44 | 17 | 31 | 24 | 18 |
| Rows/columns | 6 | 9 | 9 | 4 | 2 | 0 |
| Two circles | 86 | 77 | 77 | 38 | 51 | 45 |
| **Dataset Average** | 36.66 | 40.67 | 32 | 30.33 | 31.33 | 25.33 |

Table 10: Phi-3.5-Vision Model Performance Across Various Tasks