# OpenReview forum: "Captioning and Task-Specific Prompting for Improved VLM Performance"
_AAAI.org/2025/Workshop/NeurMAD — AAAI 2025 Workshop NeurMAD Submission_

### Official Review · Reviewer_UfCJ · 2024-12-18
**This work have the potential, but needs a lot of revamps**

**Rating:** 4
**Confidence:** 4

**Review:**

Summary:

This paper discusses the performance improvements of visual language models (VLMs) in tasks involving vision and reasoning, such as image retrieval and visual question answering (VQA). This paper proposes an efficient, question-driven image description (captioning) pipeline to enhance visual question answering in a mathematical context. The approach extracts keywords from the question, generates targeted captions for each image-question pair, and uses these captions as hints for question answering.

Strength:
- A pipeline is proposed to enhance VQA capabilities by extracting keywords from questions and generating targeted image descriptions.
- Task-specific guidance is used as a “method” to enhance the VQA and description processes.
- The robustness of the model to hostile cues was evaluated to ensure that the description-based method does not compromise too much on robustness.

Weakness:
- The Figure 1 on Page 2 is lack of the detail of model. It’s hard to find the connection between the three steps. What are their meanings? How they improved from one to another?
- In the main context, authors never mention about Figure 1-4 and Table 1-2. So What are those information related?
- In Experiment on Page 3, author mention they divide MathVision dataset into three sub-datasets. How did they work? What standard author used to split the datasets as those three parts?
- In Results on Page 3-4, What datasets are used for testing ‘Model-wise’? And Which VLM is used for testing ‘Dataset-wise’?
- Authors said their approaches are using a prompt-based pipeline to solve relative problems, which reducing the computational overhead. However, there is no any comparison we can find in this study to show the difference between authors’ approaches and other approaches.

Suggestions:
- In Figure 1 on Page 2, authors should consider to improve the text format for a batter reading.
- In Experiments on Page 3, authors mention the models they used. We suggest author add more details of each model to highlight the difference of model.
- In Result section on Page 4, ‘(Table 4)’ is actually ‘(Table 2)’.

---

### Official Review · Reviewer_cKr2 · 2024-12-20
**A nice work, but a bit premature, about VLMs capacity to perform simple boo-keeping "math" problems**

**Rating:** 4
**Confidence:** 4

**Review:**

The paper tests the performance of different VLMs in solving math problems using various zero-shot prompting techniques. The techniques themselves are nice and elaborate, as well as the variety of datasets and VLMs used. Therefore the results are convincing.

Having said that, I think that the paper is lacking in a few key aspects:
1. The VQA datasets were extensively studied in the past - I would present the performance of the best models that were trained for these tasks. This will allow us to see the gap the VLM is expected to close.
2. The paper only focuses on problems such as counting. I think that it's more interesting to ask the VLM to approximate the number of objects rather than give an exact number. A human won't be able to "see" the number of objects as well. The human observer will have to count, using his finger, which is a completely different task than seeing the image and "counting" how many items there are.
3. Use RAG, few-shot - if you want to simulate something more similar to a human process.
4. There is not enough description of the dataset, types of questions, and error analysis, so it's hard to understand why the model was wrong when he was wrong (or right).

---

### Official Review · Reviewer_fLTw · 2024-12-28
**Preliminary efforts to improve VLM performance in mathematical reasoning and visual understanding tasks**

**Rating:** 4
**Confidence:** 3

**Review:**

Summary:
This paper presents an innovative approach to improving the performance of Vision-Language Models (VLMs) in mathematical reasoning and visual understanding tasks. The authors propose a task-specific captioning pipeline that extracts keywords from the question, generates targeted image captions, and uses these captions as prompts to guide the VLM in solving complex problems. The approach is evaluated across multiple datasets and demonstrates promising improvements in accuracy and robustness.

Key Contributions:
* Generates targeted captions using keywords extracted from questions and integrates these captions into the reasoning process.
* Evaluate the pipeline's performance against adversarial approaches to ensure reliability.
* Tests on datasets involving geometry, counting, algebra, and mathematical reasoning to assess generalizability.

Strengths:
* The proposed captioning pipeline improves VLM performance on mathematical and reasoning tasks compared to baseline methods.
* By generating targeted captions and providing task-specific guidance, the method encourages VLMs to focus on visual content, improving their reasoning abilities.
* The pipeline demonstrates consistent improvements across multiple datasets and tasks, including geometry, counting, and algebra.

Limitations:
* The study evaluates only Vision-Language Models (VLMs) and does not compare its approach to other state-of-the-art methods outside the VLM domain, which could provide a more comprehensive understanding of its effectiveness.

* The experiments are confined to smaller datasets and open-source models, which limits the generalizability of the findings to larger-scale, state-of-the-art VLMs such as GPT-4.

* Generating captions from the query introduces potential challenges, as the quality of the captions depends on accurate keyword extraction and relevance to the question. Poorly structured queries or ambiguous keywords could lead to suboptimal performance.

* The study does not provide an analysis of the errors made by the framework. Identifying and categorizing these errors—whether they arise from the captioning process, reasoning steps, or question formulation—could offer critical insights for refining the approach.

The paper would benefit from additional refinement and improvements before it is ready for publication.

---

### Decision · Program_Chairs · 2024-12-30

**Decision:**

Reject

**Comment:**

 We agree with the opinions of  the reviewers.